# Eminent Antimicrobial Peptide Resistance in *Zymomonas mobilis*: A Novel Advantage of Intrinsically Uncoupled Energetics

**DOI:** 10.3390/antibiotics13050451

**Published:** 2024-05-15

**Authors:** Reinis Rutkis, Zane Lasa, Marta Rubina, Inese Strazdina, Uldis Kalnenieks

**Affiliations:** 1Institute of Microbiology and Biotechnology, University of Latvia, LV-1004 Riga, Latvia; zane.lasa@lu.lv (Z.L.); rubina.marta@lu.lv (M.R.); uldis.kalnenieks@lu.lv (U.K.); 2Alternative Plants Ltd., LV-1007 Riga, Latvia

**Keywords:** antimicrobial peptides, minimal inhibitory concentrations, *Zymomonas mobilis*, uncoupled growth, Langmuir–Blodgett compression isotherm, chemiosmotic coupling

## Abstract

Relative to several model bacteria, the ethanologenic bacterium *Zymomonas mobilis* is shown here to have elevated resistance to exogenous antimicrobial peptides (AMPs)— with regard to both peptide bulk concentration in the medium and the numbers of peptide molecules per cell. By monitoring the integration of AMPs in the bacterial cell membrane and observing the resulting effect on membrane energy coupling, it is concluded that the membranotropic effects of the tested AMPs in *Z. mobilis* and in *Escherichia coli* are comparable. The advantage of *Z. mobilis* over *E. coli* apparently results from its uncoupled mode of energy metabolism that, in contrast to *E. coli*, does not rely on oxidative phosphorylation, and hence, is less vulnerable to the disruption of its energy-coupling membrane by AMPs. It is concluded that the high resistance to antimicrobial peptides (AMPs) observed in *Z. mobilis* not only proves crucial for its survival in its natural environment but also offers a promising platform for AMP production and sheds light on potential strategies for novel resistance development in clinical settings.

## 1. Introduction

The ethanologen *Zymomonas mobilis* is a facultatively anaerobic alpha-proteobacterium, with an efficient ethanol fermentation pathway, based on Entner–Doudoroff glycolysis in combination with pyruvate decarboxylase and alcohol dehydrogenase activities [1]. It has exceptionally high ethanol productivity, exceeding that of the yeast Saccharomyces cerevisiae by several times [2]. This is primarily because the catabolism of *Z. mobilis* is loosely coupled to the energy demands of its anabolism [3] and proceeds at an excessive rate, channeling up to 98% of substrate carbon to ethanol. Although *Z. mobilis* is an obligately fermentative bacterium, able to grow only on glucose, fructose and sucrose, it also possesses an active, constitutive aerobic respiratory chain. In line with *Z. mobilis’s* uncoupled mode of metabolism, its respiratory chain also operates with low energy-coupling efficiency. It does not contribute to oxidative ATP synthesis and biomass yield in aerobically growing culture [4,5,6,7], yet it performs the rapid regeneration of intracellular NAD(P)+ pools [8]. These characteristics of *Z. mobilis* physiology are relevant to its applications in biotechnology and have attracted the interest of researchers over several decades. Successful attempts have been made to engineer its catabolic pathways and to design novel bioprocesses [9,10], also involving co-cultivations with other producer microorganisms [11,12]

However, co-cultivations with *Z. mobilis* often prove to be problematic. Early work on conjugation with *E. coli* [13], co-cultivations with yeast [14], the successful use of *Z. mobilis* biomass for treating infections [15,16,17] and other related evidence (for a review, see [18]) strongly point to antagonistic interactions between *Z. mobilis* and a wide variety of other microorganisms. It was suggested that this bacterium might be producing some unidentified antimicrobial substances (‘zymocins’). Lima et al. hypothesized that it might be an antimicrobial peptide [19]. We could not find evidence in support of the peptide nature of *Z. mobilis’s* antimicrobial activity. Yet, we identified acetate and propionate as the principal antimicrobial compounds of this bacterium. We showed that the increase in antimicrobial activity by the respiratory metabolism of aerobic cultures was paralleled by the accumulation of both acids, while the non-respiring mutant lacking the respiratory NADH dehydrogenase (ndh) or the wild-type under strictly anaerobic growth conditions did not accumulate these acids, and at the same time, had negligible antimicrobial activity [20]. In a broader sense, the ability to produce antimicrobial compounds represents just one side of the interspecies competitiveness. Another side of the coin is the resistance to inhibitors, produced by other species of the same ecological niche, including bacteria, fungi and plants.

Although *Z. mobilis* might not be synthesizing antimicrobial peptides (AMPs) of its own, resistance to those coming from other organisms might well add to its competitiveness and ability to suppress its neighbors. A priori, there seems to be some rational basis to expect resistance to AMPs in bacteria with uncoupled energy metabolism. Since the bacterial cell membrane is the key target of most AMPs [21], which function as pore-forming agents that increase transmembrane permeability, one might speculate that interference with membrane chemiosmotic energy coupling plays a central role in their antimicrobial effects. Accordingly, the survival of a bacterium like *Z. mobilis*, relying completely upon substrate-level ATP generation during ethanol fermentation, could be less vulnerable to the disruption of its membrane energy coupling. Early studies on *Z. mobilis* catabolism have already demonstrated that up to a half of the cellular protein of this bacterium is represented by its glycolytic enzymes, ensuring the high metabolic rate needed for sufficient energy supply via substrate-level phosphorylation [22,23]. In the present work, we actually demonstrate the substantially higher resistance of *Z. mobilis* to selected AMPs than that of *E. coli, S. aureus*, and several other bacteria, and discuss the relation of this finding to the uncoupled energy metabolism, as well as its potential biotechnological relevance.

## 2. Results

### 2.1. Comparative Screening of Z. mobilis MIC Values for the Selected AMPs

A comparative evaluation of the resistance against 11 AMPs revealed that *Z. mobilis* possessed significantly higher resistance than other bacteria that we had examined earlier [24], including conventional pathogens such as *P. aeruginosa* and *S. aureus* (Table 1).

The antimicrobial peptide names, corresponding sequences, size, theoretical secondary structure net charge at pH7 and the origins of peptide sequences are represented in Appendix A. 

We noted, however, that there was a substantial difference between the specific cultivation conditions of *Z. mobilis* and the rest of the tested bacteria: *Z. mobilis* required a higher concentration of inoculum. A higher amount of biomass introduced at the start of cultivation might mitigate the antimicrobial effect of AMPs, since this decreases the number of peptide molecules available to penetrate the membrane of each individual cell. Indeed, it was shown that at least 107 bound peptide molecules per bacterial cell are needed to kill it [25]. To examine if it was the large inoculum of *Z. mobilis* (and, hence, the too-small number of peptide molecules per cell) that caused the observed elevated resistance to AMPs, we compared the *Z. mobilis* resistance to that of other microorganisms by normalizing the added amounts of AMPs to the respective inocula size. For these experiments, we chose one Gram-positive and one Gram-negative model strain (*S.aureus* and *E.coli*, respectively) and two AMPs, RP556 and R10. As the cells of all three bacterial species have a roughly comparable size, the calculation of specific inhibitory concentrations for all strains was conducted by using the *E. coli* cell size parameters [26]. Normalized inhibitory concentrations of antimicrobial peptides RP556 and R10 are presented in Table 2.

### 2.2. Confocal Microscopy Examination of Intracellular Localization of the FITC-Labeled AMPs

Confocal microscopy images for *E. coli* and *Z. mobilis* showed similar results for both bacteria (Figure 1). Taken together, our examination of FITC-labeled AMPs interacting with the *E. coli* and *Z. mobilis* membranes unveiled the ability of labeled AMP to spontaneously insert into phospholipid membranes. At the same time, it is worth noting that the utilization of FITC fluorescent tags in the confocal microscope experiments was based on assumption that the tagging process did not affect the physiochemical characteristics of the respective peptides. This might well not have been true. To address this potential concern, we delved deeper into the interactions between these AMPs and membranes by employing the Langmuir–Blodgett monolayer technique.

### 2.3. The AMP Effect on Langmuir–Blodgett Compression Isotherms of Model Membranes

By providing a controlled and well-defined experimental platform, the Langmuir–Blodgett (LB) monolayer technique can yield valuable information about the mechanisms of interaction of AMPs with biological membranes. It is well established that monitoring of the surface pressure via the monolayer area, or the so called π-A isotherms of such mixed lipid/AMP Langmuir monolayers, may reveal lipid–peptide interactions [27]. To see if weak AMP–membrane interaction might be the underlying reason for *Z. mobilis’s* high AMP resistance, we applied the LB monolayer technique and compared the obtained results with our previously published data on the interaction of *E. coli* phospholipid model membranes with the same AMPs [24]. In *E. coli*, as a model Gram-negative bacteria, the membrane lipid composition is well characterized [28]. Its inner membrane is composed mainly of phosphatidylethanolamine (PE), phosphatidylglycerol and cardiolipin (CL). *Z. mobilis’s* membrane phospholipid composition is less well studied. According to [29], it contains less phosphatidylethanolamine but is enriched with phosphatidylserine and phosphatidylcholine (Table 1). Of these phospholipids, phosphatidylcholine is the less negatively charged one and, at the same time, it is more commonly found in eukaryotic cell membranes (accordingly, the eukaryotic cell membranes carry less negative charge and thus are less susceptible to positively charged AMPs).

Analysis of the π-A isotherms confirmed that both peptides, RP556 and R10, intercalated in the *Z. mobilis* phospholipid model membrane and slightly reduced its π–collapse values (Figure 2). The same effect was observed previously with an *E. coli* phospholipid model membrane [24].

To quantitatively assess the interactions within membrane monolayers, we utilized a parameter commonly employed in LB techniques: the limiting area per molecule (Aπ→0). This value is derived by extrapolating the linear segment of a densely packed solid monolayer’s π-A isotherm to zero surface pressure. In our context, Aπ→0 signifies the limiting area occupied by each membrane-forming unit, and this linear segment typically falls within the surface pressure range of 25–40 mN/m. As the surface pressure (π) of 30 mN/m generally represents the lipid packing density of a cell membrane’s outer leaflet [30], the results obtained from our model system hold biological relevance. Consistent with our findings in the *E. coli* membrane model system [24], the present results with *Z. mobilis* membranes show that within the lipid layer, R556 occupies a larger area than R10, resulting in a higher Aπ→0 value (see Table 3).

Altogether, confocal microscopy and the LB experiments confirm the membranothropic behavior of both peptides in *Z. mobilis*. Our study thus strongly indicates that the *Z. mobilis* membrane composition does not prevent its interaction with AMPs and apparently cannot serve as an explanation for the elevated AMP resistance of this bacterium.

### 2.4. AMP Uptake by Cells

While it has been established that the tested AMPs interact with bacterial membranes, the inherent amphiphilic nature of AMPs still raises questions about the proportion of added AMPs that remain in the aqueous solution, whether in the buffer system or cultivation medium. This information holds significance as it can aid in more accurately estimating the precise number of molecules required to reach the critical threshold for pore formation in biological membranes. To assess this, we introduced FITC-labeled R10 peptide to thermally inactivated *E. coli* and *Z. mobilis* cells at room temperature. Subsequently, we monitored the time course of fluorescence changes between the buffer, where the initially labeled peptides were introduced, and the inactivated cells (Figure 3). Thus, after integration in bacterial membranes, peptides transfer the fluorescence signal from the medium to the cells.

As seen from the steep increase in the fluorescence ratio, AMP integration in cell membranes occurs within the first seconds of the experiment, capturing the vast majority of added AMPs. Of course, rapid integration in membranes does not necessary mean that the final pore conformation is arranged, yet it highlights the integration speed and proportions of peptide residing in the cell membranes versus the fraction remaining in the aqueous media. In all experiments, the increase in fluorescence ratio cells/ buffer was close to 5. This observation indicates that the previously obtained normalized peptide inhibitory concentrations per cell might largely correspond to the membrane-integrated AMP fraction (even bearing in mind that, to some extent, integration in the membrane itself might alter the fluorescence).

### 2.5. The Interference of AMPs with ATP Synthesis by Artificially Induced Transmembrane pH Gradient

All the evidence accumulated by the above experiments demonstrates the interaction of the peptides with membranes of both bacteria. The key question, however, is to what extent the AMPs interact specifically with the cytoplasmic membrane in each bacterium and whether that affects energy coupling. In particular, this is relevant for *Z. mobilis*, since this Gram-negative bacterium is known for the low permeability of its outer membrane. In principle, an AMP-impermeable outer membrane could be a plausible alternative explanation of *Z. mobilis*’s high resistance to the tested AMPs. To gain a direct indication of the AMPs’ interference with energy coupling in the cytoplasmic membranes, we chose to monitor ATP synthesis in response to an artificially induced transmembrane pH gradient and to examine the putative effect of AMP addition upon the membrane energy coupling, added in concentrations above and below the inhibitory threshold.

The synthesis of ATP induced by an artificial pH gradient is related to the activity of the proton-dependent ATP synthase of the bacterial cytoplasmic membrane. pH gradient-induced ATP synthesis has been previously demonstrated both in *Z. mobilis* and *E. coli* [6,31,32]. Here, we assumed that any effect the AMPs exerted on the pH-induced ATP synthesis indicated a direct interaction between the peptide and the cytoplasmic membrane. After the induction of a pH gradient of 3.5–4.0 units in cell suspensions of both *Z. mobilis* and *E. coli*, we observed an abrupt increase in the intracellular ATP concentration, reaching its maximum value within a time span between 30 sec and 1 min. The magnitude of the observed ATP concentration rise approximately corresponded to previously published values. The addition of the protonophoric uncoupler ClCCP at a 10 μM concentration caused an almost complete de-energization and a dramatic drop of the ATP yield. Notably, the AMPs, when added in amounts exceeding the threshold value of about E10+7 molecules per cell (Figure 4c,d), also largely eliminated the ATP jump after the pH pulse, although in both bacteria peptides, it already caused some steady elevation of the intracellular ATP level during incubation before the pH shift. When added in amounts below the threshold value (Figure 4a,b), peptides had no inhibitory effect on the ATP response. These results indicate that in both bacteria, these peptides de-energize the cytoplasmic membrane to a comparable degree. Yet, such an effect elicits a much stronger physiological response in *E. coli* than in *Z. mobilis.*

## 3. Discussion

In the present study, we demonstrated that the ethanologenic bacterium *Z. mobilis*, in addition to its broad-spectrum antimicrobial activity, manifests elevated resistance to exogenous AMPs. The MIC values of 11 tested AMPs for *Z. mobilis* appeared to be substantially higher than almost all MIC values of the same peptides for several other Gram-positive and Gram-negative bacteria [24], including pathogens. For two AMPs, selected from the list of eleven, we calculated the number of peptide molecules per bacterial cell needed to induce the inhibitory effect. For *Z. mobilis*, these normalized values were also high, exceeding those of *S. aureus* by approximately 10 times and those of *E. coli* even by 3 orders of magnitude.

Several experiments confirmed that the antibacterial effects of the two selected AMPs resulted from their interaction with the cell membranes in both *E. coli* and *Z. mobilis*. In model membranes composed of phospholipid mixtures, simulating, respectively, the composition of *E. coli* [24] and *Z. mobilis* (present work) cytoplasmic membranes, the integration of AMPs in the phospholipid layer was demonstrated by Langmuir–Blodgett compression isotherms. The confocal microscopy of whole cells of both bacteria using FITC-labeled AMPs clearly showed the localization of the fluorescing peptide molecules near the cell membranes. When the labeled peptides were added to suspensions of thermally inactivated bacteria, the fluorescent signal rapidly accumulated in the biomass, while, simultaneously, the fluorescence decreased in the extracellular medium, showing that the AMPs penetrated cell membranes via passive diffusion and accumulated there at high proportions.

In order to see if the AMPs also penetrated the cytoplasmic membrane (not just the outer membrane and/or the periplasmic space) and thus could directly interfere with the membrane energy coupling in both bacteria, we applied the induction of ATP synthesis by means of an artificial transmembrane pH gradient as a test assay. We assumed that forming peptide pores in the lipid bilayer [21] would elevate its proton conductance and reduce the fraction of protons crossing the cytoplasmic membrane via the ATP synthase channel in its F_o_ part, as a result, lowering the ATP yield. ATP synthesis after the pH shift was indeed strongly reduced by the peptides, indicating that the AMPs tested here interacted with cytoplasmic membranes of both *E. coli* and *Z. mobilis*. Apparently, these peptides interfered with the hemiosmotic energy coupling via the cytoplasmic membrane F1Fo-type H+-ATP synthase in both bacteria. Although the absolute value of the ATP response to an artificial pH gradient in each bacterium was different, and close to previously reported values [6,32], in both bacteria, we observed a pronounced de-energizing effect of AMPs (when added in amounts exceeding the inhibitory threshold), resembling that of the protonophoric uncoupler ClCCP. The above data suggest that the difference between *E. coli* and *Z. mobilis* with respect to AMP resistance does not result from a higher permeability barrier of *Zymomonas’s* outer membrane or from the poor integration of the peptides in the lipid bilayer of its cytoplasmic membrane, relative to *E. coli*. Rather, it reflects the specifics of the uncoupled *Z. mobilis* energy metabolism per se. In contrast to the energy metabolism of *E. coli*, energy supply in a (micro)aerobically growing *Z. mobilis* culture is based exclusively on substrate-level phosphorylation in the E-D pathway [1,6,33] Therefore, the disruption of energy coupling via the H+-ATP synthase is less damaging to its overall viability. In other words, the inherent decoupling of energy metabolism in this bacterium, coupled with its dependence on substrate-level ATP production, enhances its resilience to inhibitory compounds that target hemiosmotic energy coupling, suggesting the existence of a novel resistance mechanism against AMPs.

The high AMP resistance of *Z. mobilis* should have strong ecological relevance for this bacterium. Apart from competitive advantage against other microorganisms, this trait is essential for its survival in its natural media, namely, plants. *Z. mobilis* is a member of plant microbiota, inhabiting sugary plant materials [16,18]. Plants synthesize a vast variety of AMPs, which serve for plant defense against pathogens and also affect the plant’s own growth and development [34]. Hence, the ability to resist media containing various AMPs of plant origin might represent a significant selective advantage in this ecological niche.

Given the significant challenge of masking the potentially lethal effects of AMPs on the bacterial host in recombinant production, it becomes all the more tempting to capitalize on the novel AMP resistance exhibited by *Z. mobilis* for the overproduction of heterologous AMPs. This, however, raises several issues, the main one of which might be the relatively low cell yield of this bacterium that directly results from its inefficient energetics—which, at the same time, might be the very basis of its high AMP resistance, as we propose here. *Z. mobilis* can be regarded as a potential producer organism of AMPs, but the design of technologically feasible production may still require a lot of research.

As the development of resistance to antimicrobial compounds remains one of the most pressing challenges for healthcare systems, the identification of this novel super-resistance mechanism to antimicrobial peptides (AMPs) observed in this study holds critical importance beyond the scope of *Z. mobilis*’s physiology.

## 4. Materials and Methods

### 4.1. Strains and Cultivation

*Z. mobilis* strains, Zm6 (ATCC 29191), *E. coli* (ATCC 25404) and an *S. aureus* strain (DSM 20231) were subjected to screening for AMP resistance. *Z. mobilis* strains were routinely grown on the standard culture medium containing (per liter) 5 g of yeast extract, 1 g of KH_2_PO_4_, 1 g of (NH_4_) 2SO_4_, and 0.5 g of MgSO_4_·7H_2_O, supplemented with 40 mL of 50% glucose solution. *E. coli* was grown on LB medium, containing (per liter) 10 g of tryptone, 5 g of yeast extract, and 10 g of NaCl. *S. aureus* was grown on Mueller Hinton Broth, 21 g per 1 L, at 30 °C.

### 4.2. Design and Synthesis of Antimicrobial Peptides

Six AMP amino acid sequences used in this study were published earlier, while five were novel peptides designed by the present project team [21]. All AMPs, including the fluorescein isothiocyanate (FITC)-labeled RP556 and R10 peptides, were synthesized at a 1 mg scale by ProteoGenix in Schiltigheim, France, utilizing the solid-phase synthesis method. The purity of these peptides, assessed through mass spectroscopy and high-performance liquid chromatography, consistently exceeded 98%, while according to APPTEST calculations, the RP556, AA139, and R11 peptides were antiparallel beta sheet-structured, and all the others formed alpha helices.

### 4.3. Quantification of Antibacterial Activity

The quantification of antimicrobial resistance against various AMPs was conducted by monitoring the bacterial growth in the presence of peptides at serial dilutions. This assay was performed in 96-well plates using the Infinite^®^ M200 PRO Multimode Microplate Reader from Tecan in Maennedorf, Switzerland, as described in our recent report [19]. All bacterial strains were cultivated at 32 °C for 13–18 h at 200 rpm. Optical density measurements (λ = 550 nm) were automatically recorded at 10 min intervals. The antimicrobial resistance activity was quantified as the minimum inhibitory concentration (MIC), which represents the lowest peptide concentration capable of preventing the growth of bacterial cells.

### 4.4. Confocal Microscopy

For confocal microscopy analysis, *E. coli* and *Z. mobilis* cells were cultured to the mid-logarithmic phase, harvested by centrifugation at 4000 rpm for 5 min, and washed twice with PBS buffer. The cells, at a concentration of approximately 0.3 mg dry weight /mL, were incubated with labeled AMPs, following the methodology described previously [35]. In brief, FITC-labeled R10 peptides (2 µg/mL) were incubated with the cells for 30 min at 37 °C. Subsequently, the cells were washed twice with PBS to remove any unbound labeled peptides. The cells were then fixed with 4% paraformaldehyde for 20 min and washed again with PBS. The localization of the peptides within the bacteria was observed using a confocal laser scanning microscope, Leica DM RA-2 (Germany), equipped with a TCS-SL confocal scanning head.

### 4.5. Langmuir–Blodgett Compression Isotherms

Langmuir–Blodgett compression isotherm experiments were conducted at a controlled temperature of 30 ± 1 °C on an antivibration table using the KSV LB instrument (KSV MiniMicro, Helsinki, Finland) equipped with two movable barriers, collectively offering a total area of 273 cm². Surface pressure measurements were performed using a platinum Wilhelmy plate with a perimeter of 39.24 mm, in accordance with the methodology outlined in [25]. The process of obtaining surface pressure versus area per molecule isotherms involved a computer-controlled incremental compression technique. The monolayer of the membrane was compressed by a defined area increment (225 mm² in this case) and then allowed to relax. The relaxation process was considered complete when the changes in surface pressure became smaller than 0.02 mN/m per second. At this point, the equilibrium surface pressure of the monolayer (π) was determined. This incremental compression–relaxation process was repeated until all isotherms were recorded and the monolayer eventually collapsed. Herein, for the study of interactions between antimicrobial peptides (AMPs) and phospholipids, bacterial model membranes were used. In *E. coli*, as a model of Gram-negative strains, according to [27], the inner membrane is mainly composed of phosphatidylethanolamine (PE), while *Z. mobilis’s* membrane composition is more diverse, with less PE content [28]. Detailed model membrane compositions used in this study are presented in Table 4.

To investigate AMP–membrane interactions, 50 μL of model membranes at the concentration of 0.2 mg protein /mL were gently deposited onto the surface of deionized water (27.2 MΩ·cm) using a Hamilton micro syringe. In experiments involving AMPs, a solution containing 5 μL of the peptide dissolved in ethanol (at the concentration of 1 mg/mL) was mixed with the model membranes before deposition on the water surface. After allowing complete solvent evaporation, measurements of surface pressure (π) versus mean molecular area (Mma) isotherms were taken in 10 min intervals. The interactions between AMPs and the membranes were characterized using the collapse pressure (π collapse) that indicated the surface pressure at which the monolayer became densely packed, and further pressure increase was restricted. This collapse led to the formation of a condensed monolayer state, limiting the membrane area per unit (Aπ→0) as π approached 0 mN/m. To derive the Aπ→0 values, the linear portion of the π-A isotherms (within the π range of 25–40 mN/m) was approximated by the linear function π = Mma * x + b, where ‘Mma’ and ‘b’ are fitting constants, and then the limiting area, corresponding to π = 0, was calculated using the equation Aπ→0 = −b/a.

### 4.6. ATP Synthase Activity Induced by Artificial pH Gradient

ATP synthase activity induced by an artificial pH gradient was measured in non-growing starved cell suspensions. For the preparation of non-growing cell suspension, cells were harvested in the late exponential phase, sedimented, washed, and resuspended in 100 mM potassium phosphate buffer (pH 7.0), to a biomass concentration of 1.5–2.0 mg dry wt /mL. Artificial transmembrane pH gradients of 3.5–4.0 pH units were induced 4 h after cell starvation on an orbital shaker (120 rpm) by the addition of 16 μL of 1 M HCl to a 200 μL aliquot of cell suspension (with 3.8–4.0 mg dry wt /mL biomass concentration). For ATP determination, 20 μL samples were quenched in ice-cold 10% TCA and assayed by the standard luciferin–luciferase method [7] using a TECAN Infinite 200Pro microplate reader (Tecan, Männedorf, Switzerland). To evaluate the impact of AMPs and the protonophoric uncoupler ClCCP upon ATP synthesis, 5 μL of peptide (10 mg/mL) or ClCCP (at a 10 μM final concentration) was added to 200 μL of starved cell suspension, 2 h (for peptides) or 5 min (for ClCCP) prior to the pH shift by HCl.

### 4.7. Uptake of FITC-Labeled AMPs in Inactivated Cells

To quantify the uptake rate of AMPs in the inactivated cells, FITC-labeled peptides were used. For this, *E. coli* DH5α and *Z. mobilis* ZM4 cells were grown to the midlogarithmic phase, harvested by centrifugation 5000 rpm for 5 min, and washed with sterile dH_2_O. The bacterial cells were resuspended in 800 µL of sterile dH_2_O and inactivated by heating for 15 min at 85 °C. Then, 7.2 mL of dH2O was added to reach optical density OD600 1, and 1 mL of the sample was taken as the control for the fluorescence measurements. Afterwards, FITC-labeled R10 peptide (5 µg/mL) was added. Then, 1 mL samples were taken at 0.5, 1, 2.5, 5, 7.5, and 10 min and centrifuged immediately at 13 000 rpm for 1 min to separate cells from the supernatant. The cells were resuspended in 1 mL of dH_2_O for fluorescence measurements. The FITC fluorescence measurements for the cells and supernatant were conducted with a FluoroMax^®^-3 (HORIBA Jobin Yvon Inc., Edison, NJ, USA) spectrofluorometer at λ_ex_ = 490 nm and λ_em_ = 530 nm.

## 5. Conclusions

Compared to several model bacteria, *Z. mobilis* demonstrated significantly elevated resistance to various AMPs, both in terms of peptide concentration in the medium and the number of peptide molecules per cell. This resistance extends beyond conventional pathogens like *E. coli* and *S. aureus*, which suggests a unique defense mechanism possessed by *Z. mobilis.* By examining the integration of AMPs into the bacterial cell membrane and their impact on membrane energy coupling, it was revealed that *Z. mobilis,* despite its similarities with *E. coli* in membrane interaction, possesses a distinct advantage due to its uncoupled mode of energy metabolism. Unlike *E. coli*, *Z. mobilis* does not rely on oxidative phosphorylation, rendering it less susceptible to disruptions in energy coupling caused by AMPs. The high resistance of *Z. mobilis* to AMPs not only contributes to its survival in its natural environment, particularly in sugary plant materials, but also presents opportunities for biotechnological applications. *Z. mobilis* could serve as a platform for AMP production due to its resistance properties, offering insights into potential strategies for combatting antimicrobial resistance in clinical settings. In principle, *Z. mobilis’s* remarkable resistance might offer a platform for microbial AMP production and also shed light on the mechanisms of novel resistance development in clinical settings. For that, however, extensive further research is still needed to address these challenges and harness the full potential of *Z. mobilis* in AMP biotechnological production.

Overall, the study sheds light on the intricate interplay between microbial physiology, membrane interactions, and antimicrobial resistance mechanisms, underscoring the importance of interdisciplinary approaches in addressing challenges related to antimicrobial resistance and microbial biotechnology.

## Figures and Tables

**Figure 1 antibiotics-13-00451-f001:**
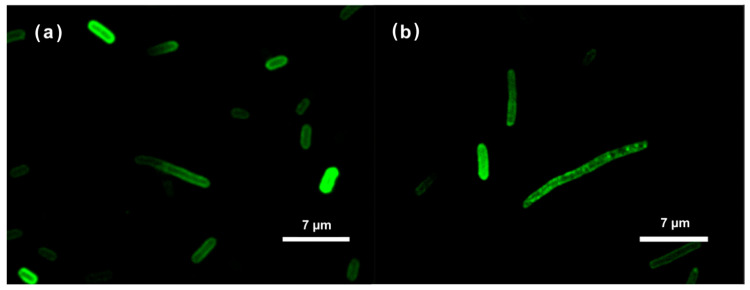
Localization of the FITC-labeled RP556 peptide in *E. coli* (**a**) and *Z. mobilis* (**b**) cells, imaged by confocal microscopy.

**Figure 2 antibiotics-13-00451-f002:**
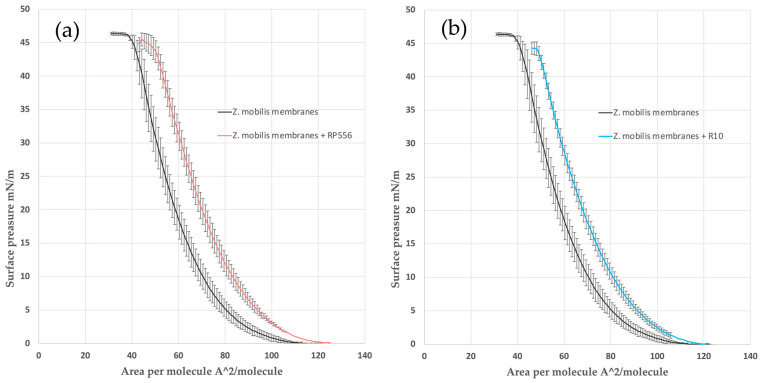
The π-A isotherms for Z. mobilis model mixtures with peptides R2 (**a**) and R10 (**b**). Dashed lines denote the linear part of the π-A isotherms, taken for the calculation of the limiting area per molecule (A_π→0_).

**Figure 3 antibiotics-13-00451-f003:**
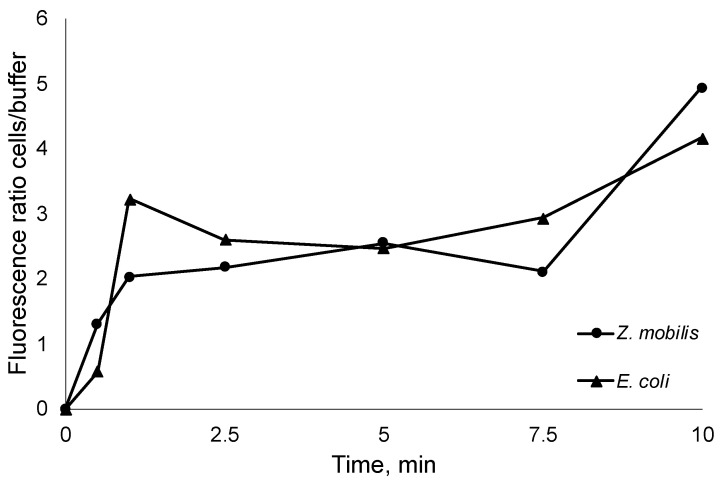
Time course of the *E. coli* and *Z. mobilis* fluorescence ratio between the thermally inactivated cells and extracellular medium.

**Figure 4 antibiotics-13-00451-f004:**
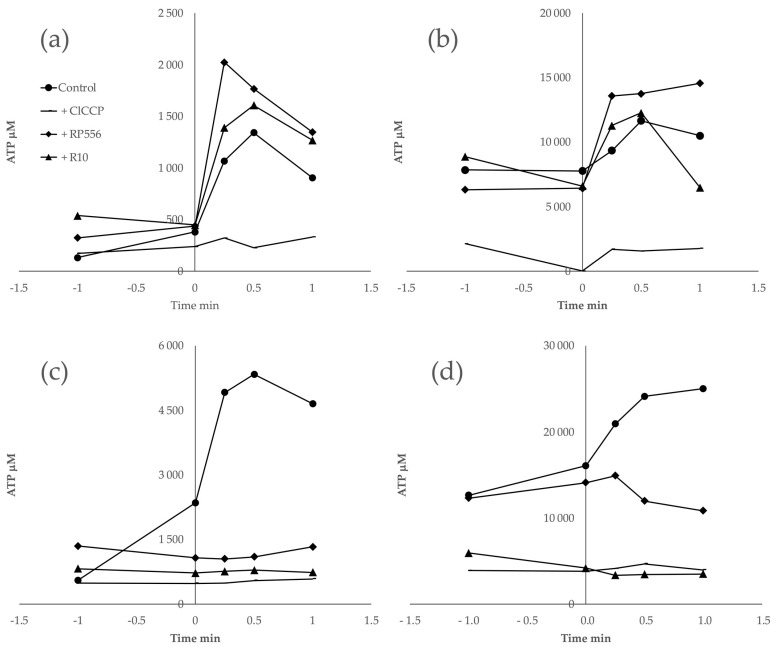
Synthesis of ATP induced by an artificial pH gradient. In (**a**,**b**), peptides were added at the concentration of 6.5 × 10^6^ molecules per cell. In (**c**,**d**), peptides were added at the concentration of 1.27 × 10^8^ molecules per cell.

**Table 1 antibiotics-13-00451-t001:** Minimal inhibitory concentrations (MICs) of AMPs against selected bacteria, μg/mL.

	MIC μg/mL							
	This Work	From [22]						
Name	*Z. mobilis*	*E. coli*	*E. Cancerogenus*	*P. aeroginosa*	*K. pneumonia*	*E. faecium*	*S. aureus*	*P. acne*
RP551	63.0	8.0	16.0	8.0	4.0	16.0	8.0	2.0
RP556	250.0	4.0	4.0	2.0	4.0	2.0	1.0	2.0
LZ1	63.0	31.0	16.0	8.0	16.0	4.0	2.0	8.0
AA139	250.0	4.0	8.0	16.0	8.0	16.0	8.0	63.0
PA13	63.0	31.0	16.0	16.0	16.0	4.0	2.0	16.0
Oligo10	31.0	16.0	63.0	63.0	63.0	16.0	4.0	2.0
R10	31.0	2.0	16.0	2.0	8.0	4.0	4.0	0.5
R11	250.0	125.0	63.0	63.0	63.0	16.0	16.0	0.3
R12	63.0	8.0	31.0	4.0	31.0	16.0	8.0	0.5
R13	63.0	4.0	31.0	4.0	8.0	16.0	4.0	0.3
R14	63.0	16.0	16.0	4.0	4.0	8.0	8.0	1.0

**Table 2 antibiotics-13-00451-t002:** Normalized inhibitory concentrations of antimicrobial RP556 and R10 peptides against selected bacteria. Values represent peptide molecules per cell needed to inhibit bacterial growth.

Bacteria	RP556	R10
*E. coli*	2.93 × 10^7^	2.30 × 10^7^
*S. aureus*	2.43 × 10^7^	1.98 × 10^8^
*Z. mobilis*	1.68 × 10^10^	6.58 × 10^9^

**Table 3 antibiotics-13-00451-t003:** Aπ→0 and π-collapse values of Z. mobilis phospholipid model membrane with peptides, derived from π-A isotherms.

Sample	Mma, Å/molecule	π Collapse, mN/m
*Z. mobilis membranes*	70.6	46.4
*Z. mobilis membranes* + RP556	85.3	45
*Z. mobilis membranes* + R10	81.3	44.2

**Table 4 antibiotics-13-00451-t004:** Phospholipid composition of the bacterial model membranes used in this study.

	Gram− [28]	*Z. mobilis* [29]
Phosphatidylethanolamin	77%	50%
Phosphatidylglycerol	13%	26%
Cardiolipin	10%	7%
Phosphatidylcholin		10%
Phosphatidylserine		7%

## Data Availability

The original contributions presented in the study are included in the article and Appendix A; further enquiries can be directed to the corresponding authors.

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
