# Peer review of "Eminent Antimicrobial Peptide Resistance in Zymomonas mobilis: A Novel Advantage of Intrinsically Uncoupled Energetics"

_antibiotics, 2024, doi:10.3390/antibiotics13050451_

Round 1

Reviewer 1 Report

Comments and Suggestions for Authors

This paper is dealing with an antimicrobial peptide (AMP) resistance in Zymomonas mobilis strain. I hove some comments about this paper.

-                      In Methods, authors define MIC as “minimum inhibitory concentration as the lowest peptide concentration capable of preventing the growth of bacterial cells”. I think authors should also compute MBC, “minimum inhibitory biocide” concentration to compare different antimicrobial peptides.

-                      It would be recommended that primary sequence and some physicochemical properties of AMPs used in previous publications should be displayed in the paper or in supplementary information.

-                      I think authors should work with a constant inoculum concentration to avoid some discussions or mistake in the interpretation of results about antimicrobial resistance of Z. mobilis or differences among different evaluated microorganisms. In this sense, I think authors should include microbial growth curves with and without AMPs in a supplementary information.

-                      I think values in Table 2 should be expressed in terms of base 10 and not with E+. The same comment in legend of Figure 4.

-                      Legends in axis Y in Figure 4 (a to d) regarding to Molar concentration should be specified. Symbol is mM? or microM? I think symbol should be corrected.

-                      Authors argue that AMPs affect cell permeability, membrane and ATP synthesis, its resistance to AMPs could be Z.mobilis is less vulnerable to disruption of its energy-coupling membrane by AMPs, and energy production in these bacteria is based mainly on substrate-level phosphorylation in the Entner-Doudoroff pathway. However, I think authors should support this with some enzymatic determinations of kinases implied in this phosphorylation.

In consequence, I think this paper can be accepted with minor revision, and additional information should be added to the text.

Author Response

1. Minimum inhibitory concentrations (MICs) represent the lowest concentration of an antimicrobial that halts visible growth of a microorganism following overnight incubation. On the other hand, minimum bactericidal concentrations (MBCs) denote the lowest antimicrobial concentration that prevents growth after subculture onto antibiotic-free media. While MICs are predominantly utilized in diagnostic laboratories to confirm resistance, they serve as a crucial research tool for assessing the in vitro efficacy of novel antimicrobials. Furthermore, data derived from MIC studies often inform the establishment of MIC breakpoints.Although MBC determinations offer additional insights, they are conducted less frequently, with their values typically aligning with MIC findings. Despite acknowledging the potential value of MBC data, we have opted not to include them in the manuscript. This decision stems from the requirement of conducting repeated experiments, which may extend the timeline and scope of the study beyond its current parameters.

2. Table with the antimicrobial peptide names, corresponding sequences, size, theoretical secondary structure net charge at pH7, and origin is included in supplementary materials.

3.  Given that multiple concentrations of a peptide are introduced into the medium, along with the examination of several strains at varying inoculum concentrations, the resultant growth curves would inundate the publication. To mitigate this, we have included a representative growth curve in the supplementary materials. This curve illustrates the impact of peptide R10 on the growth of Z. mobilis, E. coli, and S. aureus. By presenting MIC values and this representative curve, we effectively convey the peptide's effects across different microbial strains while conserving space in the main body of the manuscript.

4. Changes were made in accordance with reviewers suggestion.

5. Micro Molar (mM) concentrations are specified in all Figure 4. charts in accordance with reviewers suggestion.

6. Aready early studies on Z. mobilis catabolism demonstrated that up to a half of the cellular protein of this bacterium is represented by its glycolytic enzymes, ensuring the high metabolic rate needed for sufficient energy supply via substrate-level phosphorylation. We have added the appropriate statement and references supporting this hypothesis.

Reviewer 2 Report

Comments and Suggestions for Authors

The manuscript by Rutkis et al. reports high resistance to antimicrobial peptides observed in Z. mobilism, that may be critical for its survival in its natural environment. The manuscript is written in a simple and straightforward manner and the results are clearly and concretely presented.

Some comments:

- In line 305, authors claim: " While, according to APPTEST calculations, RP556, AA139 and R11 peptides where antiparallel beta sheet-structured, all the others formed alpha helices." Such results should be presented in the manuscript or in the supplementary information. Also, authors can comment on the relevance of this affirmation, regarding the relationship between beta-sheet and alpha helices AMP activity.

- Authors claim that findings "offer a promising platform for AMP production and sheds light on potential strategies for novel resistance development in clinical settings". Such a claim should be moderated or concretely proven.

Author Response

  1. We have included structural information in the supplementary material Table S1.
  2. We have moderated this statement – please see the part 5. Conclusions of the revised manuscript.